# MoveAnything: Controllable Scene Generation with Text-to-Image Diffusion Models

## Abstract

Controllable scene generation, *i.e.*, the task of generating images in which objects are at specific locations, is an active research area with a wide range of applications. Although Generative Adversarial Networks (GAN) have shown some successful results at this task by devising intermediate representations in which spatial content is disentangled, the quality of the generated images and the mid-level control they offer remains limited. Diffusion models, on the other hand, have been able to generate images with an unprecedented level of quality, but their generation process is hard to control and GAN-based techniques are not directly applicable to them. In this work, we propose **SceneDiffusion**, a framework that optimizes spatially disentangled representations for diffusion models. Our method jointly denoises multiple scene layouts during diffusion sampling, allowing controllable scene generation with any off-the-self text-to-image diffusion model. The proposed approach is training-free, has negligible time overhead, and is agnostic to any specific denoiser architecture. In addition, it further enables in-the-wild spatial image editing, allowing us to move any object in any given image while keeping the scene consistent. We build a comprehensive benchmark to quantitatively and qualitatively evaluate our approach and show that it outperforms previous works by a large margin on image quality and layout consistency.

## 1 Introduction

Controllable scene generation, *i.e.*, the task of generating images in which objects can be placed at specific locations, is an important topic of generative modeling (Ohta et al., 1978; Yang et al., 2021) with many commercial applications. Such applications range from content generation and editing for social media platforms to post-production of visual effects in movies and video games.

In the GAN era, latent spaces have been designed to offer a mid-level control on generated scenes (Epstein et al., 2022; Wang et al., 2022; Niemeyer & Geiger, 2021; Xu et al., 2023). Such latent spaces are optimized to provide a disentanglement between scene layout and appearance in an unsupervised manner. For instance, BlobGAN (Epstein et al., 2022) uses a group of splattering blobs for 2D layout control, and GIRAFFE (Niemeyer & Geiger, 2021) uses compositional neural fields for 3D layout control. Although these methods provide good control of the scene layout, they remain limited in the quality of the generated images and the spatial content disentanglement they offer to the user. On the other hand, diffusion models have recently shown unprecedented performance at the text-to-image (T2I) generation task (Ho et al., 2020; Dhariwal & Nichol, 2021; Rombach et al., 2021; Saharia et al., 2022). Still, they cannot provide fine-grained spatial control due to the lack of mid-level representations stemming from their fixed forward noising process (Sohl-Dickstein et al., 2015; Ho et al., 2020).

In this work, we propose a framework to bridge this gap and allow for controllable scene generation with any pretrained T2I diffusion model. Our method, entitled *SceneDiffusion*, is based on the core observation that spatial-content disentanglement can be obtained during the diffusion sampling process by denoising multiple scene layouts at each denoising step. More specifically, at each SceneDiffusion step $t$, we optimize a scene representation by first sampling several scene layouts, running locally-conditioned denoising on each layout in parallel, and then analytically optimize the representation for the next diffusion step $t - 1$ to minimize its distance with each of the locally denoised layouts. We employ a layered scene representation (Isola & Liu, 2013; Lu et al., 2020;

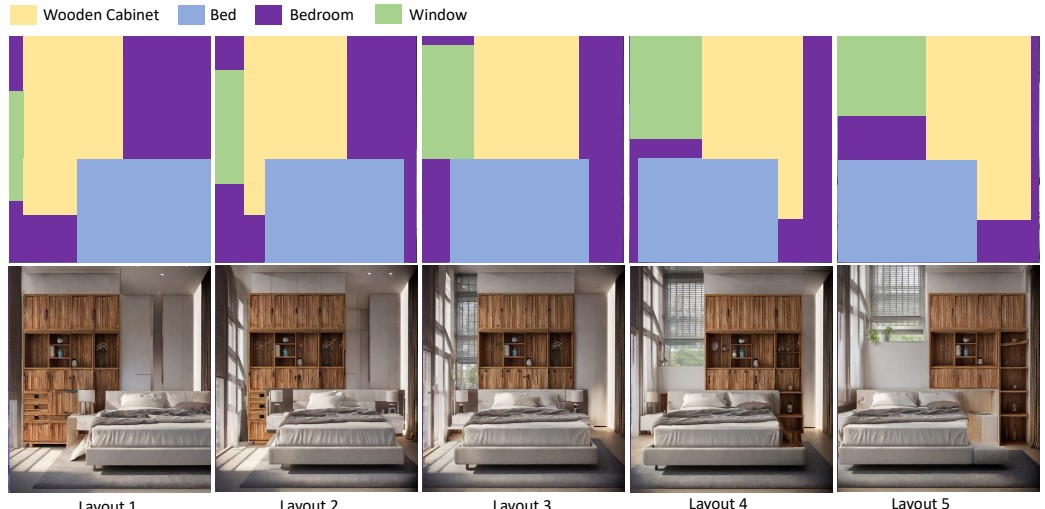

Figure 1: Our SceneDiffusion framework can generate scene images from a given layout. When the layout varies, it produces consistent scenes accross frames. SceneDiffusion enables us to precisely *move* multiple objects (*e.g.*, the bed and the window) in the image and produce a coherent output.

Kasten et al., 2021), where each layer represents an object with its shape controlled by a mask and its content controlled by a text description, allowing us to compute object occlusions using depth ordering, which further provides an in-depth understanding of objects in the generation process. Overall SceneDiffusion has negligible time overhead, does not require finetuning on paired data (Zhang & Agrawala, 2023; Mou et al., 2023b), mask-specific training (Rombach et al., 2021), or test-time optimization (Poole et al., 2022; Wang et al., 2023), is agnostic to any specific denoiser architecture and is able to generate scenes in which objects can be moved freely while keeping the contents visually consistent.

Although this approach can produce consistent scenes in which objects can be moved freely, it cannot be employed to edit objects' spatial locations in a reference image directly. To address this, given a global text prompt for the reference image, we propose to additionally use the sampling trajectory of the reference image as an anchor in the scene denoising process. When denoising multiple layouts simultaneously, we increase the weight of the reference layout in the noise update to keep the scene's faithfulness to the reference content. By disentangling the spatial location and visual appearance of the contents, our approach better reduces hallucinations compared to other baseline methods. An example application of our proposed framework is shown in Figure 1.

As one of the first works to study diffusion-based controllable scene generation and spatial image editing, we build an evaluation benchmark by creating a dataset containing 1,000 text prompts and approximately 5,000 images, with initial noise, image caption, local descriptions, and mask annotations. We evaluate our proposed approach on this dataset and show that it outperforms prior works on both image quality and layout consistency metrics by a clear margin when editing the spatial location of objects in images.

In summary, our contributions are:

- We propose a novel sampling strategy entitled *SceneDiffusion* that introduces a mid-level spatial control in the diffusion sampling process with a negligible time overhead.

- We show how our sampling strategy allows for controllable scene generation with any off-the-shelf T2I diffusion models and spatial image editing of a reference image.

- We build a comprehensive evaluation benchmark to quantify the performance of our method on the scene generation and the spatial image editing tasks in an open-vocabulary setting and show that it beats current state-of-the-art approaches by a large margin.

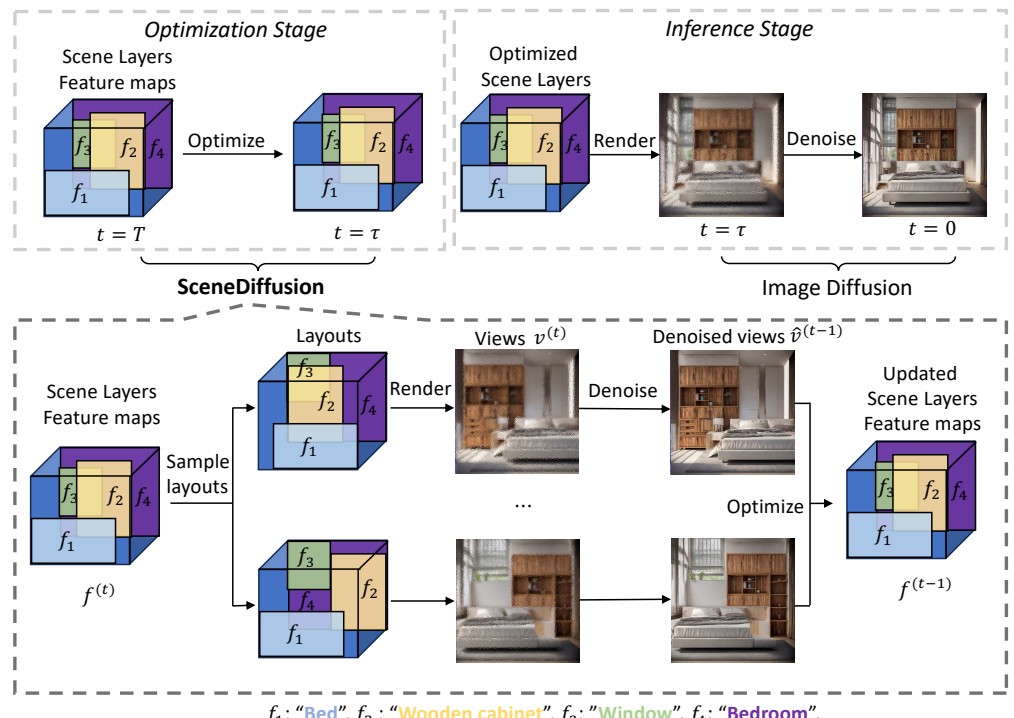

$f_1$: "Bed". $f_2$: "Wooden cabinet". $f_3$: "Window". $f_4$: "Bedroom".

Figure 2: **Method overview. (Upper)** The SceneDiffusion framework first optimizes a scene representation in the early steps of the diffusion sampling process which is then used as an input to the final image diffusion steps. **(Lower)** SceneDiffusion. SceneDiffusion updates the layer representation of an image through multiple denoising steps. Each layer representation consists of a layout *i.e.*, a set of binary masks, to present different instances, and also latent instance maps, $f^{(t)}$, for the visual appearance of each mask. In the illustration $f^{(t)}$ contains 4 features maps, from $f_1$ to $f_4$. In each denoising step, we randomly sample $N$ layouts from a predefined range and render these layouts via the current latent instance maps. Note that although this rendering process leads to $N$ latent scene images with noise, we show them in the pixel space as $v^{(t)}$ for a better understanding. We then denoise the latent scene images using a pretrained diffusion model for one step to get the latent images, $\hat{v}^{(t-1)}$, which are used to optimize the latent instance maps $f^{(t-1)}$ in the layer-representation.

## 2 RELATED WORKS

### 2.1 TEXT-TO-IMAGE DIFFUSION MODELS

Recently, diffusion models have demonstrated unprecedented results on text-to-image generation (Ho et al., 2020; Nichol et al., 2021; Dhariwal & Nichol, 2021; Rombach et al., 2021; Saharia et al., 2022), i.e., the task of generating an image from a textual description, by learning to progressively denoise an image from an input standard Gaussian noise. In the literature, T2I models vary with different design choices, including generation in pixel space (Saharia et al., 2022) or latent space (Rombach et al., 2021) and different denoiser architectures including U-Net (Ronneberger et al., 2015)-based (Ho et al., 2020) or transformer (Vaswani et al., 2017)-based (Peebles & Xie, 2022). Unlike previous image editing approaches that leverage attention cues (Hertz et al., 2022; Tumanyan et al., 2023; Chefer et al., 2023; Epstein et al., 2023) or feature correspondence (Mou et al., 2023a; Tang et al., 2023; Shi et al., 2023), our approach is agnostic to the specific design choice of the denoiser.

### 2.2 CONTROLLABLE SCENE GENERATION

Generating controllable scenes has been an important topic in generative modeling (Ohta et al., 1978; Yang et al., 2021) and has been extensively studied in the GAN context (Epstein et al., 2022; Wang

et al., 2022; Niemeyer & Geiger, 2021; Xu et al., 2023). Various approaches have been developed on applications that include controllable image generation (Epstein et al., 2022; Wang et al., 2022), 3D-aware image generation (Niemeyer & Geiger, 2021; Chan et al., 2022; Xu et al., 2023; Hong et al., 2023) and controllable video generation (Menapace et al., 2021). Usually, control at the mid-level is obtained in an unsupervised manner by building a semantically meaningful latent space. However, such techniques are not directly applicable to text-to-image diffusion models. Indeed, T2I models employ a fixed forward process (Sohl-Dickstein et al., 2015; Ho et al., 2020), which constrains the flexibility of learning a spatially disentangled latent representation, thus making it difficult to build a mapping from the desired mid-level control to the latent space. In this work, we solve this issue by optimizing a layered scene representation during the diffusion sampling process.

## 2.3 Image-editing with Diffusion Models

Off-the-shelf T2I diffusion models can be powerful image editing tools. With the help of inversion (Song et al., 2020b; Mokady et al., 2023) and subject-centric finetuning (Ruiz et al., 2023; Gal et al., 2022), various approaches have been proposed to achieve image-to-image translation including concept replacement and re-stylization (Meng et al., 2021; Hertz et al., 2022; Tumanyan et al., 2023; Kawar et al., 2023). However, these approaches are restricted to editing in local regions around objects, and editing the spatial location of objects has been rarely explored. Moreover, many of the approaches exploit an attention correspondence (Hertz et al., 2022; Tumanyan et al., 2023; Chefer et al., 2023; Epstein et al., 2023) or a feature correspondence (Tang et al., 2023; Shi et al., 2023; Mou et al., 2023a) with the final image, making the approach dependent to a specific denoiser architecture. To the best of our knowledge, this paper is one of the first to tackle spatial editing while being agnostic to any specific model architecture.

## 3 Our Approach

**Framework Overview.** Our approach is closely related to prior works on diffusion models and locally conditioned diffusion, which we briefly introduce in Section 3.1. We then discuss our sampling-based controllable scene generation approach in Section 3.2, which works towards spatial and content disentanglement for off-the-shelf text-to-image diffusion models. Leveraging this sampling approach, we can disentangle the spatial dependency for any in-the-wild image, and thus achieve spatial image editing as discussed in Section 3.3.

## 3.1 Preliminary

**Diffusion Models.** Diffusion models (Ho et al., 2020) are a type of generative model that learn to generate data from a random input noise. More specifically, given an image from the data distribution $x_0 \sim p(x_0)$ a *fixed* forward noising process progressively adds random Gaussian noise to the data, hence creating a Markov Chain of random latent variable $x_1, x_2, ..., x_T$ following:

$$q(x_t|x_{t-1}) = \mathcal{N}(x_t; \sqrt{1 - \beta_i}x_{t-1}, \beta_t \mathbf{I}), \tag{1}$$

where $\beta_1, ...\beta_T$ are constants corresponding to the noise schedule chosen so that for a high enough number of diffusion steps $x_T$ is assumed to be a standard Gaussian. We then train a denoiser $\theta$ that learns the backward process, *i.e.*, how to remove the noise from a noisy input (Ho et al., 2020). At inference time, we can sample an image by starting from a random standard Gaussian noise $x_T \sim \mathcal{N}(0; \mathbf{I})$ and iteratively denoise the image following the Markov Chain, i.e., by consecutively sampling $x_{t-1}$ from $p_\theta(x_{t-1}|x_t)$ until $x_0$:

$$x_{t-1} = x_t - \frac{1 - \lambda_t}{\sqrt{1 - \bar{\lambda}_t}}\epsilon_\theta(x_t, t) + \frac{1 - \bar{\lambda}_{t-1}}{1 - \bar{\lambda}_t}\beta_t \mathbf{z}; \quad \mathbf{z} \sim \mathcal{N}(0, \mathbf{I}), \tag{2}$$

where $\bar{\lambda}_t = \prod_{s=1}^t \lambda_s$, $\lambda_t = 1 - \beta_t$. Note that the diffusion sampling process can be accelerated by various sampling techniques as described in (Song et al., 2020b; Lu et al., 2022).

**Locally Conditioned Diffusion.** Various approaches have been proposed to generate partial image content based on dedicated text prompts using pretrained text-to-image diffusion models (Bar-Tal et al., 2023; Po & Wetzstein, 2023). For $K$ local prompts $\mathbf{y} = \{y_1, y_2, ..., y_K\}$ and binary non-overlapping masks $\mathbf{m} = \{m_1, m_2, ...m_K\}$, Locally conditioned diffusion (Po & Wetzstein, 2023)

proposes to first predict a full image noise $\epsilon_\theta(x_t, t, y_k)$ for each local prompt $y_k$ with classifier-free guidance (Ho & Salimans, 2022), and then assign it to its corresponding region masked by $m_k$:

$$\epsilon_\theta^{\text{LCD}}(x_t, t, \mathbf{y}, \mathbf{m}) = \sum_{k=1}^{K} m_k \odot \epsilon_\theta(x_t, t, y_k), \tag{3}$$

where $\odot$ is element-wise multiplication. In our approach, we employ a similar locally conditioned noise estimation for each sampled layout.

## 3.2 CONTROLLABLE SCENE GENERATION

Given a list of ordered object masks and their corresponding text prompts, we would like to generate a scene where object locations can be changed on the spatial dimensions while keeping the image content consistent and high quality. We leverage an off-the-shelf T2I diffusion model that generates in the image space (or latent space) $I \in \mathbb{R}^{c \times w \times h}$, where $c$ is the number of channels and $w$ and $h$ the width and height of the latent image, respectively. To achieve controllable scene generation, we introduce a layered scene representation in Section 3.2.1 for mid-level control and propose a new sampling strategy in Section 3.2.2 for optimization.

### 3.2.1 LAYERED SCENE REPRESENTATION

We decompose a controllable scene into $K$ layers $[l_k]_{k=1}^K$, ordered by the depth of the objects. Each layer $l_k$ has *1)* a fixed object-centric binary mask $m_k \in \{0,1\}^{c \times w \times h}$ (*e.g.*, a bounding box or segmentation mask) to show the geometric property of the object, *2)* a two-element offset, $o_k \in [0; \mu_k] \times [0; \nu_k]$, indicating its spatial locations, with $\mu_k$ and $\nu_k$ defining the horizontal and vertical movement range, and *3)* a latent feature map $f_k^{(t)} \in \mathbb{R}^{c \times w \times h}$ representing its visual appearance at diffusion step $t$.

We name the masks and their associated offsets as a *layout*. The masks $m_k$ can be generated automatically from segmentation models (Kirillov et al., 2023), or from user input, as illustrated in Section 4.3. The offset $o_k$ of each layer can be randomly sampled from the movement range $[0; \mu_k] \times [0; \nu_k]$. Note that we set the last layer $l_K$ as the background so that $m_K = \{1\}^{c \times w \times h}$ and $o_K = [0, 0]$. Besides, the latent visual map $f_k^{(T)}$ is independently initialized from a standard Gaussian noise $\mathcal{N}(0, I)$ across all the layers. We update $f_k^{(t)}$ in a sequential denoising process, detailed in Section 3.2.2.

When a layered presentation for the scene is determined, we can *render* it to a single latent scene image and then decode the latent into the pixel space. Similarly to prior works in controllable scene generation (Epstein et al., 2022) and video editing (Kasten et al., 2021), we use $\alpha$-blending to composite all the layers. More concretely, the latent scene image $v$, denoted as *view*, can be calculated as:

$$v^{(t)} = \sum_{k=1}^{K} \alpha_k \odot \overline{\text{move}}(f_k^{(t)}, o_k); \quad \alpha_k = \overline{\text{move}}(m_k, o_k) \prod_{j=1}^{k-1} (1 - \overline{\text{move}}(m_j, o_j)). \tag{4}$$

Each element in $\alpha_k \in \{0,1\}^{w \times h}$ indicates that the visibility of that location in the $k$-th latent feature map, and the function $\overline{\text{move}}(\cdot, o)$ means that we spatially shift the values of the feature map $f$ or mask $m$ by $o$. The rendering process can be applied to the layer presentation at any diffusion step, resulting in a latent scene image with a certain noise level. Mathematically, since $\alpha$ is binary and $\sum_{k=1}^{K} \alpha_k^2 = 1$, it can be proven that the rendered latent image from the initial layer presentation still follows the standard Gaussian distribution. This allows us to denoise directly on the latent images using pretrained diffusion models and update the layer presentation accordingly.

### 3.2.2 GENERATING SCENES IN DIFFUSION SAMPLING

We propose *SceneDiffusion* to optimize the latent feature maps in the layer representation from Gaussian noise. Given a layered representation, each SceneDiffusion step renders multiple latent scene images denoted as views, estimates the noise from the views, and updates the latent feature maps from the estimation.

Specifically, SceneDiffusion samples $N$ groups of offset $[o_{1,n}, o_{2,n}, \cdots, o_{K,n}]_{n=1}^N$, with each offset $o_{k,n}$ being an element of the movement range $[0; \mu_k] \times [0; \nu_k]$. This leads to $N$ layout variants from the original one. Note that different sampling methods are compared in Section 4.3, and we observe that uniform sampling in a legitimate region works the best. Meanwhile, a higher number of layouts helps the denoiser locate a better mode while also increasing the computational cost, as shown in Section 4.3. From the $K$ latent feature maps, we render the layouts as $N$ latent images $v_n \in V$:

$$V^{(t)} \equiv \{v_1^{(t)}, ..., v_N^{(t)}\}; \quad v_n^{(t)} = \sum_{k=1}^K \alpha_k \odot \overline{\text{move}}(f_k^{(t)}, o_{k,n}) \tag{5}$$

Each view $v_n^{(t)}$ follows a standard Gaussian noise, allowing us to use them as input into pretrained T2I diffusion models. Finally, we stack all the view images in each SceneDiffusion step and predict the noise $\{\hat{\epsilon}_n\}_{n=1}^N$ using locally conditioned diffusion (Po & Wetzstein, 2023) described in Equation 3:

$$\hat{\epsilon}_n^{(t)} = \epsilon_\theta^{LCD}(v_n^{(t)}, t, \mathbf{m}, \mathbf{y}), \forall n \in \{1, 2, \cdots, N\} \tag{6}$$

where $\mathbf{m}$ are the object masks, and $\mathbf{y}$ are local text prompts for each layer. Since we can run multiple layout denoising in parallel, computing $\{\hat{\epsilon}_n\}_{n=1}^N$ brings little time overhead, while costing an additional memory consumption proportional to $N$. We then update the views $v_n^{(t)}$ from the estimated noise $\hat{\epsilon}_n^{(t)}$ using Equation 2 to get $\hat{v}_n^{(t-1)}$.

Since each view corresponds to a different layout and is denoised independently, conflict can happen in overlapping mask regions. Therefore, we propose to optimize each latent feature map $f_k^{(t-1)}$ so that the rendered view from Eq 5 is close to denoised views by optimizing the following objective:

$$f^{(t-1)} = \underset{\tilde{f}^{(t-1)}}{\arg\min} \sum_{n=1}^N ||\hat{v}_n^{(t-1)} - \sum_{k=1}^K \alpha_k \odot \overline{\text{move}}(\tilde{f}_k^{(t-1)}, o_{k,n})||_2^2 \tag{7}$$

This least square problem has a closed-form solution that we obtain by updating $f$ with a weighted average of the estimated noise on each view, as shown in Eq. 8. Note that we use $\overline{\text{move}}(x, -o)$ to denote the values in $x$ translated in the reverse direction of $o$. The derivation for this solution is similar to the discussion in Bar-Tal et al. (2023).

$$f_k^{(t-1)} = \frac{\sum_{n=1}^N \overline{\text{move}}(\alpha_k \odot \hat{v}_n^{(t-1)}, -o_{k,n})}{\sum_{n=1}^N \overline{\text{move}}(\alpha_k, -o_{k,n})}, \quad \forall k \in \{1, \cdots, K\} \tag{8}$$

### 3.2.3 IMAGE DIFFUSION WITH SCENEDIFFUSION

Despite improving consistency across different views, using SceneDiffusion througout the full diffusion process leads to degraded visual quality especially in lighting and shadowing. To mitigate this side effect, we use SceneDiffusion only in the early stage of the diffusion process and then run vanilla image sampling for $\tau$ steps in the later stage. We find the resulting images to be more natural and aligned with text prompts while preserving consistency. Adjusting the value of $\tau$ can control the consistency-quality trade-off. According to our ablation in Sec. 4.3, a value of $\tau$ in 25% to 50% of the total diffusion steps strikes the best balance.

### 3.3 APPLICATION TO IMAGE EDITING

One advantage of our SceneDiffusion is that we keep an implicit expression of scene views during diffusion, which allows further controls on the views. For instance, SceneDiffusion can be conditioned on a given image as an anchor view, allowing us to change the layout of an existing image, an essential task in image editing. Concretely, when a scene image is given along with an existing layout, we take its latent feature map as the optimization target final state of an anchor view, denoted as $\hat{v}_a^{(0)}$. Then, we add Gaussian noise to this view at the diffusion noise levels, creating a sequence of anchor views for the target as different denoising steps.

$$\hat{v}_a^{(t)} = \sqrt{1 - \beta_t} \hat{v}_a^{(0)} + \beta_t \epsilon; \quad \epsilon \sim \mathcal{N}(0, 1), \forall t \in [1, \cdots, T]. \tag{9}$$

In each sampling step, we use the corresponding anchor view $\hat{v}_a^{(t)}$ to further constraint $f^{(t-1)}$, which leads to an extra weighted term in Eq. 7, as shown in Eq. 10.

$$f^{(t-1)} = \underset{\tilde{f}^{(t-1)}}{\arg\min} \sum_n w_n ||\hat{v}_n^{(t-1)} - \sum_{k=1}^{K} \alpha_k \overline{\text{move}}(\tilde{f}_k^{(t-1)}, o_{k,n})||_2^2; w_n = \begin{cases} w & \text{if } n = a, \\ 1 & \text{otherwise.} \end{cases} \quad (10)$$

Here $n \in \{1, \cdots, N\} \cup \{a\}$, $o_{k,a}$ can be computed from the layout of the given image, and $w$ controls the importance of $\hat{v}_a^{(t)}$. The closed form solution of this equation is similar to Eq. 8 and can be found in Appendix.

## 4 EXPERIMENTS

### 4.1 EXPERIMENTAL SETUP

**Dataset.** We curate a dataset of 5,092 generated images associated with image captions, object masks, local descriptions for each mask, and the initial diffusion latent noise. This is achieved by first generating 1k image captions using a Large Language Model (LLM). The LLM is instructed to describe only a single subject, in a single scene, in the following format: "[CAPTION] Subject: [SUBJECT]. Scene: [SCENE]." in order to provide prompts for local regions (see appendix for examples of such prompts). Then, we employ a $512 \times 512$ publicly available text-to-image diffusion model to generate ten images for each caption using the deterministic DDIM (Song et al., 2020a) sampler. This process results in a set of 10k images with their corresponding initial random noise. Finally, we employ an off-the-shelf open-vocabulary segmentation model Kirillov et al. (2023) to generate a subject mask for each image and apply a rule-based filter to remove low-quality images, which results in a dataset of 5,092 high-quality subject-centric images.

**Metrics.** Our main metrics for controllable scene generation are *Mask IoU*, *Consistency*, *LPIPS*, and *SSIM*. *Mask IoU* measures the layout accuracy of the generated image. Other metrics compare multiple generated images and evaluate their differences: *Consistency* for the layout, *LPIPS* for perceptual, and *SSIM* for structural changes. Moreover, in the image editing experiment, we report *FID* and *KID* to measure the similarity of the edited images to the original ones, and *CLIP score* to measure the text alignment degradation in the edited area.

Please see the Appendix for more details on our dataset construction and metrics selection.

### 4.2 IMPLEMENTATION

We implement our approach on the Diffusers library using publicly available text-to-image latent diffusion models. Two model variants are used, one employs a $64 \times 64$ latent and generates $512 \times 512$ image, and the other one employs a $128 \times 128$ latent space and generates $1024 \times 1024$ images. For classifier-free guidance (Ho & Salimans, 2022), we set the guidance scale to 7.5 for the $512 \times 512$ model and 12.5 for the $1024 \times 1024$ model. We employ the DDIM sampler (Song et al., 2020b) and the number of sampling steps is 50. We run the experiment on a single machine equipped with 8 32GB NVIDIA V100 GPUs. The total running time of a scene image generation process ranges from 5 to 30 seconds.

### 4.3 LAYOUT-CONDITIONED SCENE GENERATION

**Task.** For controllable scene generation, we evaluate our approach on the task of generating scenes where the subject mask is placed at two different spatial locations. The overall content of the scene is expected to be invariant to the mask location. For this, we start by generating a given scene using local prompts and giving a location to the object of interest. Then we apply a random shift in [-0.2, 0.2] of the image size in both vertical and horizontal directions to generate the target scene. Note that for this task, we use a subset of our dataset which consists of 100 prompts and 231 images.

**Baselines.** We compare our approach to MultiDiffusion (Bar-Tal et al., 2023). Following the same protocol, we use a 20% solid color bootstrapping strategy. Concretely, in the first 10 steps of the 50-step DDIM scheduling, we fill the area outside the object mask with a random solid color. For the initialization, instead of initializing the two target layouts with two independent random noises, we

Table 1: **Controllable scene generation.** We compare with MultiDiffusion (Bar-Tal et al., 2023) on controllable scene. †: without the solid color bootstrapping (Bar-Tal et al., 2023)

| Method | Mask IoU ↑ | Consistency↑ | LPIPS ↓ | SSIM ↑ |
|---|---|---|---|---|
| MultiDiffusion† | $0.262 \pm 0.008$ | $0.267 \pm 0.012$ | $0.517 \pm 0.009$ | $0.456 \pm 0.009$ |
| MultiDiffusion | $0.453 \pm 0.024$ | $0.426 \pm 0.011$ | $0.506 \pm 0.010$ | $0.493 \pm 0.012$ |
| Ours† | $0.298 \pm 0.017$ | $0.605 \pm 0.009$ | $\mathbf{0.192} \pm 0.003$ | $0.769 \pm 0.004$ |
| Ours | $\mathbf{0.515} \pm 0.010$ | $\mathbf{0.723} \pm 0.016$ | $0.211 \pm 0.002$ | $\mathbf{0.767} \pm 0.003$ |

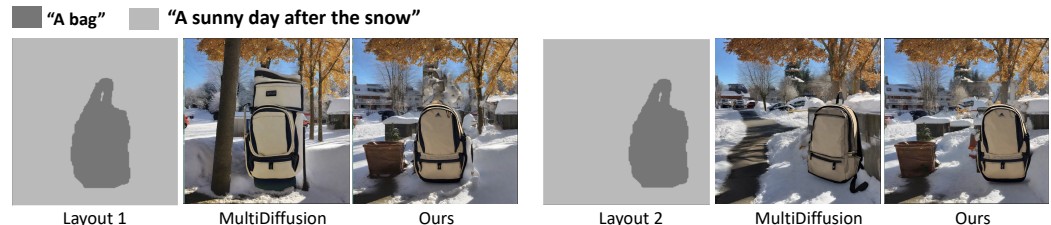

Figure 3: **Qualitative comparison for controllable scene generation.** Between layout 1 and 2 Multidiffusion (Bar-Tal et al., 2023) is able to move the backpack to the target location, but both the scene and the object change. Our method can produce coherent and consistent images with minimal visual appearance difference.

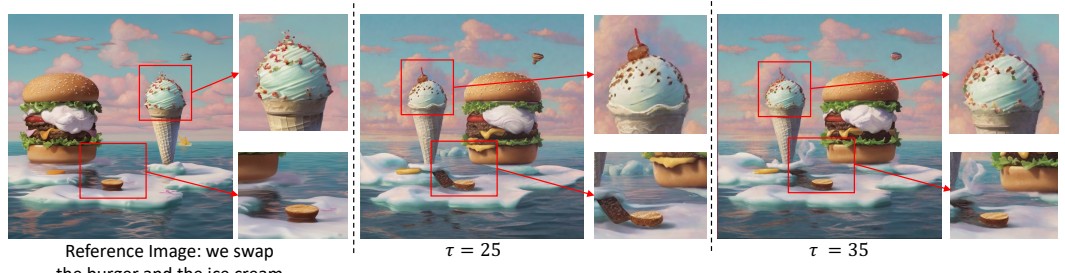

Figure 4: **Spatial Image editing with Anchored Fusion.** We swap the locations of two objects from a reference image and stop the SceneDiffusion at an early and a late step, *e.g.*, 25 *vs.* 35. Stopping SceneDiffusion at a later step tends to prevent hallucination on the objects (Right image). Prompt: *A burger and an ice cream cone floating in the ocean.*

use the layer representation to ensure the consistency between foreground and background, which leads to better cross-view consistency. For all approaches, we use a 50-step DDIM scheduler.

**Results.** We present quantitative results in Table 1. All experiments are repeated with 5 random seeds and we report the mean and standard deviation. The results show that SceneDiffusion significantly outperforms MultiDiffusion on all metrics. A qualitative comparison is enlisted in Figure 3.

### 4.4 Image-Conditioned Scene Generation: Moving Object

**Task.** For object moving, our task is to move the subject in the reference image to a new target location. This task differs from the previous controllable scene generation task in that image contents are given, and only the overall scene prompts are used. Our task is therefore to measure not only if the object content is consistent but also whether the overall scene is close to the reference image. Similar to the aforementioned procedures, we randomly sample a mask offset from [-0.2, 0.2] of image size for both horizontal and vertical directions. We use the full test dataset for evaluation.

Table 2: **Object moving comparison.** We compare with inpainting approaches on object moving in a reference image. Our method achieves the best result accross all metrics.

| Method | FID $\downarrow$ | KID $_{\times 10^3}$ $\downarrow$ | Mask IOU $\uparrow$ | Clip Score $\uparrow$ | LPIPS $\downarrow$ | SSIM $\uparrow$ |
|---|---|---|---|---|---|---|
| Inpainting | $10.267 \pm 0.020$ | $1.167 \pm 0.026$ | $0.620 \pm 0.001$ | $0.321 \pm 0.000$ | $0.278 \pm 0.001$ | $0.671 \pm 0.000$ |
| Inpainting[†] | $6.383 \pm 0.039$ | $0.099 \pm 0.014$ | $0.747 \pm 0.002$ | $0.321 \pm 0.000$ | $0.264 \pm 0.001$ | $0.680 \pm 0.001$ |
| Ours | $\mathbf{5.289} \pm 0.022$ | $\mathbf{0.059} \pm 0.014$ | $\mathbf{0.817} \pm 0.003$ | $0.321 \pm 0.000$ | $\mathbf{0.263} \pm 0.001$ | $\mathbf{0.709} \pm 0.000$ |

Table 3: **Ablation on controllable scene generation.** We compare our method by varying the number of views and SceneDiffusion steps $\tau$. †: Layout using deterministic sampling at fixed intervals.

| Views | Step $\tau$ | Mask IoU $\uparrow$ | Consistency $\uparrow$ | LPIPS $\downarrow$ | SSIM $\uparrow$ |
|---|---|---|---|---|---|
| 2 | 25 | $0.477 \pm 0.020$ | $0.619 \pm 0.017$ | $0.274 \pm 0.004$ | $0.697 \pm 0.004$ |
| 8[†] | 25 | $0.485 \pm 0.006$ | $0.638 \pm 0.011$ | $0.269 \pm 0.002$ | $0.699 \pm 0.004$ |
| 8 | 25 | $0.499 \pm 0.005$ | $0.657 \pm 0.012$ | $0.274 \pm 0.001$ | $0.689 \pm 0.004$ |
| 2 | 25 | $0.477 \pm 0.020$ | $0.619 \pm 0.017$ | $0.274 \pm 0.004$ | $0.697 \pm 0.004$ |
| 2 | 37 | $0.483 \pm 0.024$ | $0.661 \pm 0.023$ | $0.227 \pm 0.004$ | $0.753 \pm 0.003$ |
| 2 | 50 | $0.501 \pm 0.015$ | $0.699 \pm 0.019$ | $0.208 \pm 0.005$ | $0.778 \pm 0.004$ |
| 8 | 50 | $0.515 \pm 0.010$ | $0.723 \pm 0.016$ | $0.211 \pm 0.002$ | $0.767 \pm 0.003$ |

**Baselines**. We use inpainting-based approaches as our baseline. Specifically, we first crop the subject from the reference image, paste it to the target location and then inpaint the blank areas. To better blend the subject with the background, we dilate the edge of the subject for 30 pixels for all candidates. We compare our approach with two inpainting models: a standard text-to-image diffusion model using the RePaint technique (Lugmayr et al., 2022), and a specialized inpainting model trained with masking. For both inpainting models, we use the default 25-step DPM solver.

**Results.** We report quantitative results in Table 2. The experiment is repeated on 5 random seeds and the mean and standard deviation are recorded. Our approach outperforms the standard inpainting model by a large margin. Compared to a specifically fine-tuned model, our approach also demonstrates a clear advantage on all metrics. A qualitative comparison is shown in Figure 4.

## 4.5 DISCUSSION

**Ablation on $N$ and $\tau$.** In Table 3, we analyze the effect of the number of views and SceneDiffusion steps. We observe that having more views and more SceneDiffusion steps leads to a better disentanglement between the subject and the background of the scene, as indicated by higher Mask IoU and Consistency. Please see the Appendix for more ablation experiments on the image editing.

**Limitation.** Our method presents several limitations. Firstly, in the alpha compositing, we assume the masks to be binary. Our method is therefore not handling transparent objects in the scene. In addition, the object's appearance may not fit tightly to the mask in the final rendered image. Finally, our approach requires a large amount of memory to simultaneously denoise multiple layouts, restricting the applications in resource-limited user cases.

## 5 CONCLUSION

We proposed SceneDiffusion, a method capable of achieving controllable scene generation using off-the-shelf T2I diffusion models. At the core of our method, we use a pretrained diffusion model to optimize a layered scene representation during the early stages of the diffusion sampling process. Thanks to the layered representation, spatial and appearance information are disentangled which allows to generate controllable scenes. By additionally leveraging the sampling trajectory of a reference image as an anchor, we can condition SceneDiffusion on a reference image to allow for spatial image editing tasks. Compared to baseline approaches like MultiDiffusion and Inpaiting, our approach achieves significantly better generation quality and layout consistency on our proposed benchmark while having negligible time overhead.

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

# A DATASET AND METRICS

## A.1 CAPTION GENERATION

The prompt we used to generate image captions is as follows:

*Please give me 100 image captions that describe a single subject in a scene. The format is as follows: "A cat is sitting in a museum. Subject: cat. Scene: museum.". "Cat" is the subject and "museum" is the scene.*

Example image captions are as follows:

1. A bird is perched on a windowsill. Subject: bird. Scene: windowsill.
2. A goldfish swims in a bowl. Subject: goldfish. Scene: bowl.
3. A kite soars above the beach. Subject: kite. Scene: beach.
4. A bicycle leans against a brick wall. Subject: bicycle. Scene: brick wall.
5. A turtle crawls along a sandy path. Subject: turtle. Scene: sandy path.
6. A sunflower stands tall in a garden. Subject: sunflower. Scene: garden.
7. A butterfly rests on a blooming flower. Subject: butterfly. Scene: blooming flower.
8. A tree casts its shadow on a playground. Subject: tree. Scene: playground.
9. A cloud drifts over a mountain peak. Subject: cloud. Scene: mountain peak.
10. A snake slithers through the tall grass. Subject: snake. Scene: tall grass.
11. A boat floats on a serene lake. Subject: boat. Scene: lake.
12. A moon illuminates a silent forest. Subject: moon. Scene: silent forest.
13. A star shines brightly in the evening sky. Subject: star. Scene: evening sky.
14. A ladybug crawls atop a green leaf. Subject: ladybug. Scene: green leaf.
15. A guitar rests against a campfire. Subject: guitar. Scene: campfire.
16. A watch lays on a wooden table. Subject: watch. Scene: wooden table.
17. A windmill spins in the midst of a field. Subject: windmill. Scene: field.
18. A lighthouse stands sentinel over the coast. Subject: lighthouse. Scene: coast.
19. A cactus stands alone in the desert. Subject: cactus. Scene: desert.
20. A book lies open on a park bench. Subject: book. Scene: park bench.

## A.2 IMAGE FILTERING

The following filters are used in GroundedSAM (Liu et al., 2023) to remove images with no or ambiguous foreground objects when constructing the test dataset:

- No bounding box detected.
- Bounding box confidence lower than 0.5.
- Bounding box area is larger than 60% of the image size.
- Segmentation mask is smaller than 5% of the image size.

## A.3 METRICS

*Mask IoU.* We employ the segmentation model to predict the subject mask on the generated images. One of the two target layouts contains the original annotated mask. We can, therefore, compute a mask IoU between the annotated mask and the shifted mask.

*Consistency.* We compute the mask IoU between the subject masks for the two generated images. To compensate for masks that move out of the canvas, we align the masks in two different layouts respectively and take maximum IoU.

Table 4: **Object moving ablation.** We compare our method with inpainting approaches on object moving for varying number of views and SceneDiffusion steps $\tau$

| Views | Steps $\tau$ | FID $\downarrow$ | KID $\downarrow$ | Mask IOU $\uparrow$ | Clip Score $\uparrow$ | LPIPS $\downarrow$ | SSIM $\uparrow$ |
|---|---|---|---|---|---|---|---|
| 2 | 25 | $5.918 \pm 0.018$ | $-0.020 \pm 0.004$ | $0.788 \pm 0.003$ | $0.322 \pm 0.000$ | $0.294 \pm 0.001$ | $0.672 \pm 0.001$ |
| 8 | 25 | $5.890 \pm 0.032$ | $-0.010 \pm 0.004$ | $0.794 \pm 0.002$ | $0.321 \pm 0.000$ | $0.289 \pm 0.001$ | $0.676 \pm 0.000$ |
| 2 | 12 | $7.401 \pm 0.025$ | $-0.079 \pm 0.009$ | $0.667 \pm 0.003$ | $0.322 \pm 0.000$ | $0.368 \pm 0.001$ | $0.598 \pm 0.001$ |
| 2 | 25 | $5.918 \pm 0.018$ | $-0.020 \pm 0.004$ | $0.788 \pm 0.003$ | $0.322 \pm 0.000$ | $0.294 \pm 0.001$ | $0.672 \pm 0.001$ |
| 2 | 37 | $5.289 \pm 0.022$ | $0.059 \pm 0.014$ | $0.817 \pm 0.003$ | $0.321 \pm 0.000$ | $0.263 \pm 0.001$ | $0.709 \pm 0.000$ |
| 2 | 50 | $5.320 \pm 0.029$ | $0.182 \pm 0.020$ | $0.836 \pm 0.003$ | $0.322 \pm 0.000$ | $0.255 \pm 0.001$ | $0.722 \pm 0.001$ |

*LPIPS.* We compute the LPIPS distance between the two generated views to examine the cross-view perceptual consistency.

*SSIM.* We compute the SSIM similarity between the two generated views to examine the structural similarity. We compute the FID between the edited images and the test dataset to evaluate the image quality.

*KID.* Similar to FID, we report KID as well for image quality evaluation.

*Clip Score.* We measure the similarity between the image embedding and the text embedding to ensure that the text alignment does not degrade after editing.

## B    ADDITIONAL QUALITATIVE RESULTS

Additional qualitative results for the object moving task are presented in Figure 5.

## C    ADDITIONAL ABLATION EXPERIMENTS

**Ablation for image editing experiment.** In Table 4, we show that the multi-layout joint diffusion is crucial to achieve strong quantitative performance.

## D    CLOSED-FORM SOLUTIONS

**Solution to Eq. 10.** The analytical solution to this problem is:

$$f_k^{(t-1)} = \frac{\sum_n w_n \overline{\text{move}}(\alpha_k \odot \hat{v}_n^{(t-1)}, -o_{k,n})}{\sum_n w_n \overline{\text{move}}(\alpha_k, -o_{i,n})}; \quad \forall k \in \{1, \cdots, K\}. \tag{11}$$

Here $n \in \{1, \cdots, N\} \cup \{a\}$, $o_{k,a}$ can be computed from the layout of the given image.

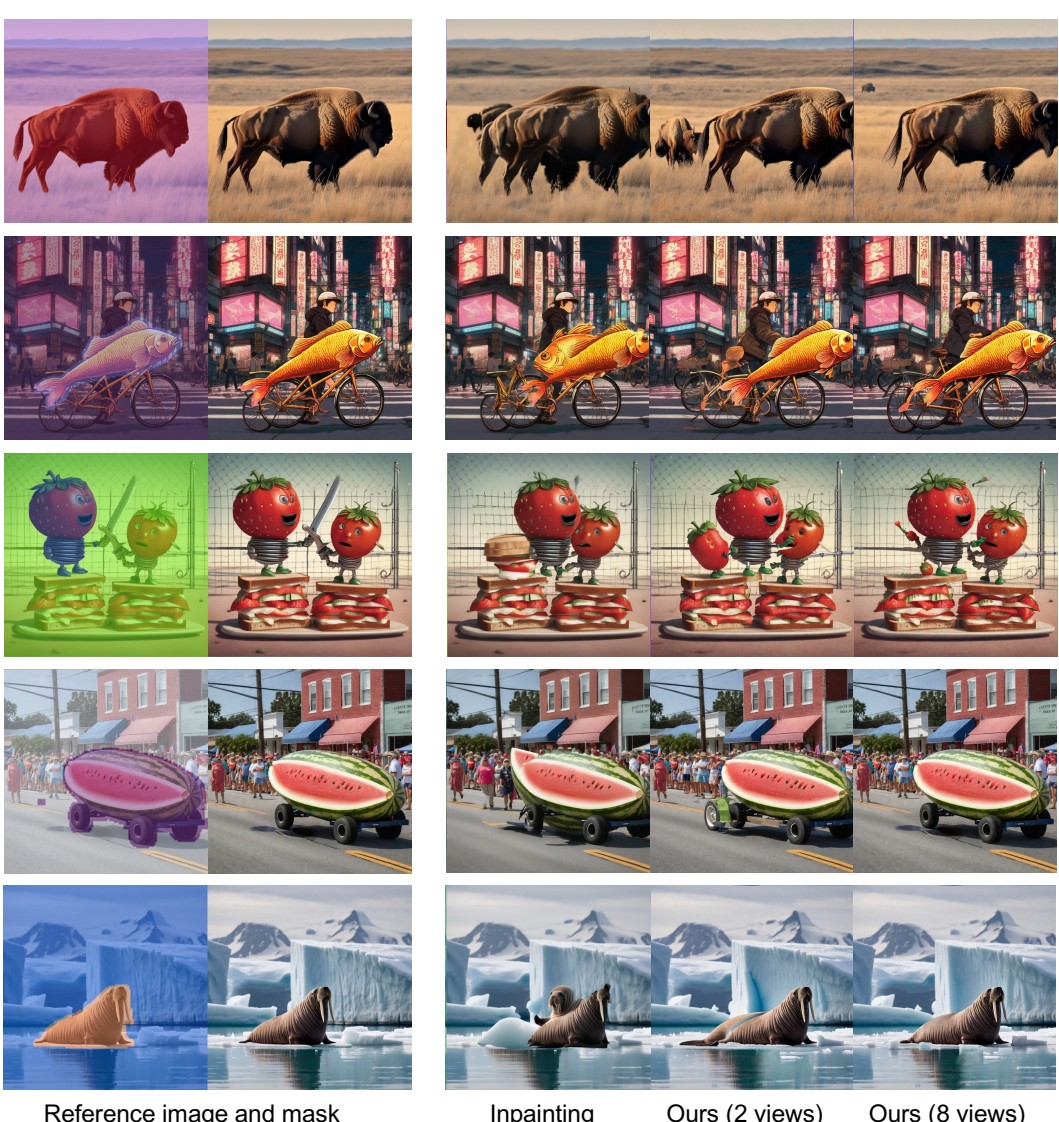

Reference image and mask      Inpainting    Ours (2 views)    Ours (8 views)

Figure 5: Qualitative comparison for object moving on the $1024 \times 1024$ diffusion model. We move the object mask to the right of the image. Inpainting methods create repetitive textures or hallucinations in the background, while our method can produce coherent and consistent images with minimal visual difference.

