# OpenReview forum: "MoveAnything: Controllable Scene Generation with Text-to-Image Diffusion Models"
_ICLR.cc/2024/Conference — ICLR 2024 Conference Withdrawn Submission_

### Official Review · Reviewer_M51x · 2023-10-29

**Soundness:** 3 good
**Presentation:** 2 fair
**Contribution:** 2 fair
**Rating:** 5
**Confidence:** 3

**Summary:**

The paper "MoveAnything: Controllable Scene Generation with Text-to-Image Diffusion Models" proposes a method for controllable scene generation compatible with any trained text-to-image diffusion model. The method builds on top of layered scene representations (LSR), where each layer in the scene corresponds to the (latent) feature representation of an object. The paper proposes to use the Segment Anything model to produce these segmentations that then correspond to layers. The core idea of the paper is to user several LSRs, randomly move/translate each of these layers in each 'view', perform a diffusion update step, and pool the updates afterwards by a weighted average. This method is analyzed on 1 novel dataset, and 2 tasks: layout conditioned scene generation, and image-conditioned moving of an object.

**Strengths:**

+ the core idea of the paper, SceneDiffusion (the term they use to denote the multi-view sampling and pooling process) is an original contribution to the literature.
+ the general field of controllable image generation is very intersting for the computer vision community, and also of great practical importance
+ the experiments that are shown indicate both a qualitative, as well as a quantitative improvement over the baselines

**Weaknesses:**

- Clarity: the idea of the paper is not well laid out in my opinion - i.e. the paper could be written significantly better. E.g. the caption of Figure 2 (the main method idea) is basically not understandable without reading the paper itself (small things: in the optimization box, nothing changes with optimization? I think you should change the features as they're updated, e.g. from f_1 to f'_1; also both Figure 1 and Figure 2 use rectangular masks, which on first reading, I took as a limitation of the paper). In both the paper and the caption the core idea should be laid out first (i.e. multiple views, but then pooling/averaging the resulting features). Further, a theoretical, or intuitiv, motivation should be given as to why this is believed to work well (after all the authors must have had something in mind when starting the study) - e.g. the averaging removes unwanted hallucinated artifacts (I'm 'hallucinating' here of course).
- The experiments are quite limited and only applied to a single dataset. Further, this dataset is generated, and not a ground-truth image dataset. It seems the method is not applicable to any real image not generated from a diffusion process? If so, this is a severe limitation that should be pointed out. Also, as the dataset is not to be published(?), this hampers scientific reproducibility.

Smaller things:
- p.4. "Diffusion Models", you cite Ho et al., which is great, but there has been work before (e.g. [1])
- p.6. 'implicit expression': why is this implicit? I'd argue it's explicit.

[1] Deep Unsupervised Learning using Nonequilibrium Thermodynamics. Jascha Sohl-Dickstein, Eric Weiss, Niru Maheswaranathan, Surya Ganguli. Proceedings of the 32nd International Conference on Machine Learning, PMLR 37:2256-2265, 2015.

**Questions:**

- Will the dataset be published for other scientists to investigate the same setting?

---

### Official Review · Reviewer_b2vZ · 2023-10-29

**Soundness:** 2 fair
**Presentation:** 2 fair
**Contribution:** 2 fair
**Rating:** 3
**Confidence:** 4

**Summary:**

This paper introduces a sampling strategy called SceneDiffusion for mid-level controllable scene generation and spatial image editing by any pretrained diffusion models without training. In the early stage of the generation process, SceneDiffusion analytically optimizes a locally -conditioned layered scene representation with a pretrained text-to-image diffusion model. This operation allows spatial and appearance information disentanglement, thus SceneDiffusion achieves controllable scene generation and spatial image editing, i.e., moving objects in an image of a scene.

**Strengths:**

1. **The motivation behind this work is strong**. Generating controllable scenes through the movement of objects within an image is a promising task.

2. **The proposed method is intriguing and effective**. This work defines the moving generative task as a least-squares problem, which has a closed-form solution. Consequently, the proposed method does not require any training.

**Weaknesses:**

1. **Contribution of the proposed method is overclaimed.**

- The authors assert that their work is "one of the first works to study diffusion-based controllable scene generation and spatial image editing". However, there have been numerous studies on these two research topics, such as [1] (controllable scene generation) and [2] (spatial image editing). The authors may change the expression to specifically refer to "moving objects." Nonetheless, there are some concurrent works, such as [3] and [4], which are not adequately discussed in the manuscript.
- The movement of objects in experiments is limited to [-0.2, 0.2] of image size, and the provided visualizations suggest that the proposed method's object movements are minor, restricting the proposed method's applications. Is it hard to significantly alter the spatial positions using the proposed method?

2. **Experiments are not sufficient and rigorous.**
- Although Figure 1 and 2 illustrate an example of moving multiple objects in a scene image, the curated benchmark solely contains images with only one object, which makes the experiments less convincing.
- Some experimental settings are unclear. For example, (1) How is the subset dataset for layout-conditioned scene generation evaluation selected? (2) Why do the proposed method and baselines use different diffusion schedulers for image-conditioned scene generation?
- $w$ is a critical hyperparameter for the image editing task, but its value is not provided and is not investigated in the ablation study.

3. **The writing style could be improved, and some presentations are confusing.**
- The symbols in some of the most critical equations, such as Equation (7) and (10), are confusing. $f^{(t-1)}$, $\tilde{f}^{(t-1)}$ and $\tilde{f}_k^{(t-1)}$ appear simultaneously. Do they have the same meaning?
- Figure 2 is misleading as it divides the generating process into the "optimization stage" and "inference stage". However, both of these stages are the inference stage for a diffusion model, and the "optimization stage" does not actually optimize network weights or input images. "Modification/modified" might be a better word than "optimization/optimized," in my opinion.

---
[1] Make-A-Scene: Scene-Based Text-to-Image Generation with Human Priors, ECCV'22

[2] RePaint: Inpainting using Denoising Diffusion Probabilistic Models, CVPR'22

[3] DragonDiffusion: Enabling Drag-style Manipulation on Diffusion Models, ArXiv'2307

[4] AnyDoor: Zero-shot Object-level Image Customization, ArXiv'2307

**Questions:**

1. **Is this work the first work to study diffusion-based scene generation and spatial image editing?** If no, please refrain from using the term "first". If yes, but there have been concurrent works (e.g., within the last four months), please include them in the related work section and discuss the differences.

2. **How to get the depth of objects, which is required to determine the order of layers?** While the experiments only investigate the single-object situation with two layers (one for the object and one for the background), does the multi-object situation presented in Figure 1 require explicit assignment of object orders by the users?

3. **How is the subset dataset for layout-conditioned scene generation evaluation selected?** It would be helpful to explain the selection process and the reasoning behind it to avoid cherry-picking.

4. **Why do the proposed method and baselines use different diffusion schedulers for image-conditioned scene generation?** Specifically, the proposed method uses a 50-step DDIM scheduler, while the baselines use a 25-step DPM solver.

5. **How do you set the value of $w$ for the image editing task?** How does this hyperparameter affect the performance of image editing?

6. **Cloud you please explain the meaning of $\tilde{f}$ in Equation (7) and (10)?** Does it share the same meaning as $f$, i.e., the latent visual map to be optimized analytically? It would be helpful to clarify its meaning in the manuscript.

---

### Official Review · Reviewer_hfBH · 2023-11-01

**Soundness:** 1 poor
**Presentation:** 3 good
**Contribution:** 3 good
**Rating:** 5
**Confidence:** 2

**Summary:**

This paper introduces a mid-level scene representation that can be used to control text-to-image diffusion models. The key idea is that one could revert locally conditioned diffusion to obtain feature maps for individual regions in the image. Given this observation, the paper designs a 2D scene representation which represents each individual objects as a mask and a corresponding latent feature map, which can be shifted along pre-defined xy ranges. By sampling multiple xy offsets, one could obtain a collection of layouts and their corresponding blended features. Applying pre-trained diffusion models to these result in multiple images corresponding to different layout variations, which can be used to update the latent feature maps for the individual objects (and using multiple images for this step increases the quality of the latent representation for the individual objects).

Evaluation shows that the proposed method outperforms previous diffusion-based work on controllable scene generation, and inpainting methods on object moving. The qualitative results look good, though only a few of them are provided.

**Strengths:**

I have not worked closely with diffusion models so this is a rather common sense review based on my very generic understandings.

-The method appears to be technically sound overall. Prior works are properly discussed and useful relevant ideas are incorporated in many parts of the work. Although the current 2D scene representation is rather rudimentary, I can see how this can be extended to allow more complex scene editing tasks.
-There is a good amount of novelty as well, though most of the techniques used seem to be traceable to early works (correct me if I am wrong here), they are combined in a clever way to allow generating consistent images across multiple layout variations.
-It is also impressive that this can be achieved with a frozen, pretrained model. The general idea of how to leverage a pre-trained model to obtain task-specific features/representations could definitely inspire other works.
-The result look convincing, outperforming relevant baselines (though I am not sure if they are the most proper baselines as I am not familiar with recent related works), and appears plausible qualitatively for the few examples shown.

**Weaknesses:**

- Although the idea is quite novel, most of the machine learning techniques used in this paper can be traced back to earlier works.
- I feel that the current setting is a bit too constrained: each mask seems to be fixed in size and can only shift in image space, this is usually not the case in real world scenarios, where objects can move in three dimensions, and undergo rotation as well. It seems to me that it would be challenging to extend the current approach to these scenarios, given that the feature map is discretized in pixel space and any other transformation will result in a change of mask size/shape.
- Following the previous point, I am rather uncertain about the assumption that objects under different layout can share a single latent feature map, which is different from a (semantic) latent code. It appears to me that there is the assumption that the object appearance will stay similar when the layout changes, however, I don't think this is usually the case, as there will be changes in perspective/lighting/occlusion/etc that accompanies layout changes. All the examples shown in the paper show basically a front view of things so this is not that noticeable, but I would really want to see the method in other cases e.g. what if the room in figure 1 is wider, and moving the bed causes the side of the bed to be visible in certain layouts? What about Figure 3, where MultiDiffusion produces a backpack in a different pose (right)? Is the proposed method also able to generate anything that is not the exact from view of the bag?  What if object in some "views" is occluded but not in other views?  These seem to be rather important cases that the method should be able to handle.
- I probably won't have these questions if the evaluation are more comprehensive. I do find the evaluation very lacking, especially the lack of qualitative samples. I think some prelim on prior techniques can be trimmed to allow more space for visualizations.

**Questions:**

As stated in the weakness section, I would lean more positive if the authors could showcase that the proposed method can extend beyond simple xy shift on the image plane, and able to handle changes in object appearances across different layout variations. Would also be helpful to include more visuals.
As mentioned earlier, I haven't worked closely with diffusion, so I might have misunderstandings here and there. Any clarifications would be greatly appreciated.

---

### Official Review · Reviewer_Ndze · 2023-11-02

**Soundness:** 2 fair
**Presentation:** 3 good
**Contribution:** 2 fair
**Rating:** 5
**Confidence:** 1

**Summary:**

Controllable scene generation is an important research problem with many potential applications. Given a pretrained Diffusion model, it is hard to directly control the layout of the generated image precisely. The paper proposes a general framework for controlling the layout of the generated images. Specifically, it proposes to render layered latent feature maps on the canvas for diffusion process and proposes two techniques to improve the generated image quality. The proposed approach is training-free, has little time overhead, and can be applied to different model architectures. It also allows the spatial image editing while keeping the scene consistent. Moreover, the authors propose a benchmark to evaluate its capabilities on controllable image generation and demonstrates better performance than previous approaches.

**Strengths:**

- the idea of leveraging layered latent feature maps for different objects is interesting
- the proposed approach is training-free, and computationally efficient for inference
- the capability for in-the-wild image-editing with consistent background between edits is a nice property\

**Weaknesses:**

- Missing an important related work GLIGEN [1].
  * It is the first work to bring layout-controllability to stable diffusion and should be compared as a baseline in layout-to-image generation.
  * It should be cited in Sec. 3.2.3 as the described mitigation is the same as the scheduled sampling in GLIGEN paper
[1] Li, et al. "Gligen: Open-set grounded text-to-image generation." CVPR 2023.

- Given the layout-to-image setting and the illustration in Fig. 1, the paper does not quantitatively evaluate on the accuracy of the bounding box conditioning. The authors are suggested to evaluate the precision of the object location on COCO dataset following the setting in [1], as the COCO dataset is more well-established and contains complex scenes as well.

- There are no quantitative experiments / qualitative results on if the proposed approach can handle overlapping regions/masks. Suppose a person is standing in front of a car, the car bbox will be split to half by the human -- the resulting rendered result is ambiguous: is this a person standing in front of a car, or it stands between the two cars. Despite the layered latent feature maps, they are rendered to the same canvas $v_n$, and the delta noise is predicted by the u-net on $v_n$, which has the ambiguity issue.

**Questions:**

Please see the weaknesses section, and I am happy to increase the rating if the concerns are addressed.

---

### Author Response · Authors · 2023-11-17
**Thank you for your reviews!**

We sincerely thank all reviewers for your constructive comments! We will withdraw the paper for now and revise it accordingly.